# MOESR: Multi-objective Evolutionary algorithm for image Super-Resolution

## Abstract

In recent years, deep neural networks have gained substantial traction in the field of super-resolution. However, existing deep learning methods primarily focus on enhancing the peak signal-to-noise ratio (PSNR) of images, resulting in suboptimal performance across various evaluation metrics and a lack of fine details in image visual quality. To address these limitations, we introduce a comprehensive algorithmic framework, Multi-Objective Evolutionary Algorithm for Image Super-Resolution (MOESR), which aims to achieve a balanced optimization of multi-objective in image super-resolution. Specifically, MOESR first decomposes the multi-objective super-resolution problem into sub-problems and employs a novel approach to generate an initial population for the evolutionary algorithm. Subsequently, it enhances mutation, crossover, and update processes using an improved differential evolution algorithm, yielding a more Pareto-efficient set of solutions. Compared to traditional gradient-based methods, our approach does not require gradient calculations for each objective. As a result, it avoids issues such as gradient vanishing or local optima. Furthermore, our method has lower computational complexity, making it particularly advantageous for addressing high-dimensional problems and deep networks. Extensive experiments are conducted on five widely-used benchmarks and two multi-objective tasks, resulting in promising performance compared to previous state-of-the-art methods. In addition, our approach can not only address multi-objective optimization problems but also represents the first method capable of addressing the balance between objective and perceptual metrics. Our code will be released soon.

## 1 Introduction

Super-resolution (SR) is a extensively studied field, aiming to transform low-resolution inputs into visually appealing high-resolution images. Its applications span across various computer vision domains, including security and surveillance imaging (Zhang et al., 2010), medical imaging (Li et al., 2021), and object recognition. An integral aspect of SR involves quantifying discrepancies between distorted images and reference images, driving research in objective image quality assessment to develop automated perceptual quality measures.

In the realm of SR evaluation metrics, several options exist. The mean squared error (MSE) and its derivative, peak signal-to-noise ratio (PSNR), are widely employed full-reference quality metrics, measuring pixel-level intensity differences between distorted and reference images. The Structural Similarity Index (SSIM) (Wang et al., 2004) assesses structural information preservation. Learned Perceptual Image Patch Similarity (LPIPS) (Back et al., 1997) links perceptual judgments to feature representations. Additionally, perceptually motivated distance metrics such as MSSIM (Wang et al., 2003), FSIM (Zhang et al., 2011), and HDR-VDP (Mantiuk et al., 2011) have been proposed.

**Why do we need the multi-objective optimization in image super-resolution?** While these metrics are valuable, current SR research primarily optimizes deep neural networks using single-target objective functions. For example, the L1 loss predominantly improves PSNR, VGG loss enhances LPIPS, and SSIM loss directly boosts SSIM. However, this focus on a single metric often compromises the performance of other metrics once neural networks converge.

Prior SR research mainly explores model structures and learning strategies to benefit all metrics, but this often leads to poor trade-offs between them, a common issue in multi-objective learning.

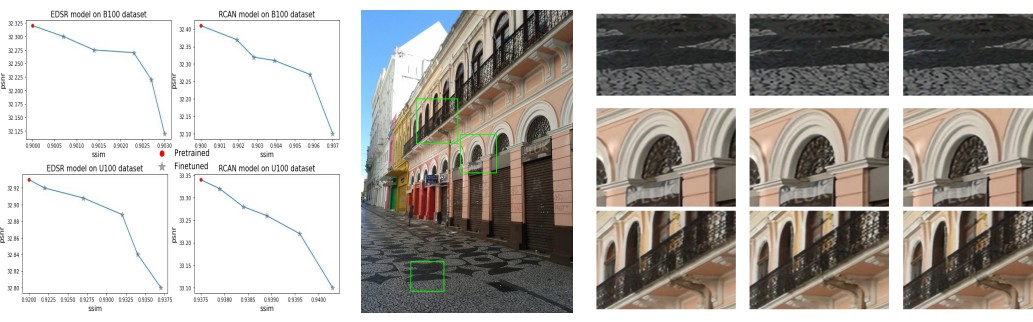

| Finetune models by combining different losses with random weights | Reference HR image | L1 loss
31.33/0.9064 | SSIM loss
31.02/0.9090 | MOESR
31.28/0.9100 |

Figure 1: The trade-off phenomenon when training super-resolution networks. Where improving the value of one single objective function during the optimization process result in the deterioration of other objective functions. Our proposed multi-object evolutionary algorithm based on decomposition provides a better trade-off result.

Therefore, considering multiple metrics when evaluating the quality of high-resolution outputs from distorted signals is advisable. As shown in Figure 1 left, networks trained with a single objective tend to prioritize one metric, and as training progresses, their performance on the other metric starts to decline. To address this problem, we propose selecting a fair SR solution from the Pareto Frontier using a fairness metric. Consequently, research on multi-objective optimization in SR tasks becomes crucial, as it tackles potential conflicts between objectives. Compared to traditional single-objective optimization methods, our approach achieves a better trade-off, as depicted in Figure 1 on the right.

**Why do we need the evolutionary algorithm?** In the realm of multi-objective optimization research, two primary approaches are commonly followed: gradient descent-based algorithms (GD) and evolutionary algorithms (EA). The GD multi-objective optimization is frequently applied in multi-task learning to handle task interdependencies and competition. However, it falls short of being a genuine multi-objective optimization technique. In contrast, EA have demonstrated strong performance in addressing multi-objective optimization challenges in numerical optimization problems. In addition, GD employs a single-point approach to iteratively search for the optimal solution. This approach excels at exploiting the solution space to find a single optimal solution. However, it performs poorly in the context of multi-objective optimization. Compared to GD, EAs employ a population-based approach to search for solutions within the multi-objective optimization space.

**What's the limitation of the current EA method?** EAs are adept at exploring a wide solution space to discover multiple diverse solutions that are distributed along the Pareto front (non-dominated front). However, In previous research, multi-objective optimization was rarely applied to neural networks, and even when applied, the network size was often limited to LeNet's scale (Gong et al., 2020). It did not demonstrate satisfactory performance for deep neural networks and big data in practical applications, such as image super-resolution. Therefore, our objective is to implement evolutionary multi-object optimization algorithms in super-resolution tasks involving deep neural networks.

**Which evolutionary algorithm we should choose?** There are many evolution algorithms such as genetic algorithms, differential evolution, particle swarm optimization and so on. We mainly use the differential evolution algorithms and step into several variants of adaptive DE, such as SADE, SHADE and so on. In addition, we provide a simple method to generate a reference point and Pareto Front, which are used to evaluate the quality of the solution set generated by multi-objective optimization algorithms.

In summary, we employ a Multi-Objective Evolutionary Super-Resolution (MOESR) framework to optimize crucial objectives for single-image super-resolution. Our MOESR is capable of handling multiple objective optimizations, such as SSIM and PSNR. It is also the first method that simultaneously addresses both objective and perceptual metrics, specifically PSNR and LPIPS. The primary goal of our MOESR is to identify Pareto-efficient solutions that consider both objectives. To achieve this, we introduce a decomposition-based evolving MO algorithm and enhance its performance by implementing SHADE as an improved differential evolution strategy. We conduct extensive experiments on five widely-recognized benchmarks and two multi-objective tasks, which yield promising results compared to previous state-of-the-art methods. Furthermore, we compare our results with

gradient-based multi-objective optimization strategies, and our solution consistently outperforms other baseline methods significantly, demonstrating that our solutions are nearly Pareto efficient.

The contributions of this work are:

- We propose a general Pareto efficient algorithmic framework (MOESR) for multi-objective image super-resolution. This framework is designed to be adaptable to various models and objectives, showcasing its impressive scalability. Importantly, it represents the first method capable of simultaneously addressing both objective and perceptual metrics.

- Building upon MOESR as the foundation, we demonstrate how to generate the Pareto Frontier and establish evaluation metrics for SR tasks. Specifically, this work is the first to propose the selection of a fair SR solution from the Pareto Frontier using appropriate fairness metrics.

- We propose an enhanced version of the differential evolution algorithm for our evolution MO super-resolution task and this is the first work evolving deep neural networks for multi-object optimization in real world applications.

- We conduct extensive experiments on benchmark datasets. The results indicate that our algorithm result in promising performance compared to previous state-of-the-art method.

## 2 RELATED WORK

### 2.1 IMAGE SUPER-RESOLUTION

Over the past decade, deep learning-based studies in super-resolution (SR) have exhibited superior performance compared to classical methods, (Yang et al., 2012) Super-resolution research encompasses various categories, including model structure exploration, multi-frame super-resolution, blind super-resolution, inference acceleration, and reference-based super-resolution. Model structure exploration involves investigating architectural designs such as attention mechanisms (Zhang et al., 2018a), residual and dense connections (Zhang et al., 2018b), non-local blocks(Zhou et al., 2020), and transformers (Liang et al., 2021), to enhance model performance. Multi-frame super-resolution (Shi et al., 2016) utilizes multiple low-resolution input images from different sources to improve the reconstruction quality of high-resolution images. Video super-resolution is a specific form of multiple-image super-resolution that applies the relationship between successive frames to enhance the resolution of the entire video sequence. Blind super-resolution (Gu et al., 2019) aims to recover high-resolution images from a single low-resolution image without prior knowledge or reference images. Inference acceleration optimizes the computational efficiency and speed of super-resolution models through techniques like lightweight architectures (Hui et al., 2019). Reference-based super-resolution (Yang et al., 2020) employs additional high-resolution images as references to enhance the reconstruction quality of low-resolution images by leveraging details and structural information. However, the study of evaluation metrics for super-resolution poses significant challenges, and there exists a paucity of research in this domain, particularly concerning the comprehensive assessment of methods using multiple metrics.

### 2.2 MULTI-OBJECTIVE OPTIMIZATION

The problem of finding Pareto optimal solutions given multiple criteria is called multi-objective optimization. A variety of algorithms for multi-objective optimization exist. One such approach is the multiple-gradient descent algorithm (MGDA) (Désidéri, 2012), which uses gradient-based optimization and provably converges to a point on the Pareto set. MGDA is well-suited for multi-task learning with deep networks. It can use the gradients of each task and solve an optimization problem to decide on an update over the shared parameters. However, there are two technical problems (Sener & Koltun, 2018) that hinder the applicability of MGDA on a large scale. (i) The underlying optimization problem does not scale gracefully to high-dimensional gradients, which arise naturally in deep networks. (ii) The algorithm requires the explicit computation of gradients per task, which results in a linear scaling of the number of backward passes and roughly multiplies the training time by the number of tasks. The alternative optimization algorithm that have been proposed in the literature, evolutionary computation (EC) has been widely recognized as a major approach for multi-objective optimization (MO). These algorithms can be divided into methods

according to dominance, index, and decomposition. Among them, Fast Non-dominated Sorting Genetic Algorithm (NSGA-II) (Deb et al., 2000), index-based Evolutionary algorithm (Das et al., 2007), and Multi-objective Evolutionary algorithm based on Decomposition (MOEA/D) (Zhang & Li, 2007) are representative algorithms, respectively. The simplest method to construct the subproblems in MOEA/D is the weighted sum method, where each subproblem is formulated as a weighted sum of the original objective functions, and the weights determine the trade-off between the objectives. The weights can be randomly assigned or generated using various techniques such as uniform distribution, Latin hypercube sampling, or random scalarization.

### 2.3 EVOLVING ALGORITHMS

Of particular relevance to our work is evolving algorithms. EA-based methods provide alternative gradient-free ways to DNN training by the metaphors of natural evolutionary processes, where a population of neural network topologies and weights evolves for better fitness globally (Stanley et al., 2019). Popular EAs algorithms for optimizing DNN include genetic algorithms (Montana et al., 1989), genetic programming (Suganuma et al., 2017), differential evolution (DE) (Pant et al., 2020), and evolution strategies (Salimans et al., 2017). However, EA-based methods were only reported to work well on small datasets and small DNNs (Piotrowski, 2014). When optimizing DNNs' weights on large-scale datasets, EA-based methods suffer from very slow (or failure of) convergence, given a large number of model parameters and a complex search space for obtaining the deep representation. Piotrowski reported the stagnation issues of several variants of adaptive DE, such as SADE, JADE, and DEGL, in optimizing network weights for regression problems (Piotrowski, 2014). In this paper, we mainly focus on the differential evolution algorithm and its variants.

## 3 PROPOSED METHOD

### 3.1 PRELIMINARY

**Multi-objective Evolutionary algorithm.** The Multi-objective Evolutionary Algorithm (MOEA) is a traditional approach that aims to aggregate different objectives into a scalarizing function using a dedicated weight vector. This process transforms Multi-objective Optimization Problems (MOP) into multiple single-objective optimization sub-problems, with each sub-problem's optimum representing a Pareto-optimal solution of the original MOP. In the context of single image super-resolution, we define the input as low-resolution (LR) images denoted as $x$ and the ground truth high-resolution (HR) images denoted as $y$. The SR tasks' objective is to train a neural network capable of generating higher-resolution images solely from the low-resolution input $x$.

### 3.2 MOESR

In order to better enhance the SR problem, we introduced a multi-objective optimization algorithm into this problem. Specifically, we let $f_1(x), f_2(x), \ldots, f_m(x)$ represent the $m$ objective functions, where $x$ is the vector of decision variables. Hence, our **objective function** is:

$$\min G(x) = (f_1(x), f_2(x), \ldots, f_m(x))^T$$

$$x = (x_1, x_2, \ldots, x_n)^T$$

where the decision vector $x$, are n dimensional factors. And this need to satisfy

$$g_j(x) \leq 0, \quad \text{for } j = 1, 2, \ldots, p$$

$$h_k(x) = 0, \quad \text{for } k = 1, 2, \ldots, q$$

**Pareto Dominance** (Deb, 2011) could promote efficient resource allocation and get the best trade-off results. It fosters fairness by ensuring no individual is disadvantaged without benefiting others. It is defined as: given two solutions $A$ and $B$, $A$ dominates $B$ ($A \prec B$) if and only if:

$$f_i(A) \leq f_i(B) \quad \text{for all } i = 1, 2, \ldots, m$$

$$f_j(A) < f_j(B) \quad \text{for at least one } j \text{ in } 1, 2, \ldots, m$$

Hence, the solution $x^*$ is said to be **Pareto optimal** if there is no other solution that dominates it. And the **Pareto front** is the set of all Pareto optimal solutions in the objective space.

However, due to the multi-objective optimization problem is too difficult and does not easily weigh the individual objectives, we decompose the objective function by this:

$$F(x) = \lambda_1 f_1(x) + \lambda_2 f_2(x) + \ldots + \lambda_m f_m(x)$$
$$st \sum_i \lambda_i = 1 \tag{1}$$

where the symbols $\lambda_1, \lambda_2, \lambda_3, \ldots, \lambda_m$ are non-negative weights assigned to each objective function.

For our super-resolution problem, when given a batch of paired low-resolution images ($x$) and high-resolution images ($y$), our approach begins by initializing a population of individuals within a solution space. This population serves as the initial set of solutions. Next, we decompose the objective function into multiple single-objective functions and create a set of decomposition weight vectors. These weight vectors guide the search direction for each single-objective optimization subproblem.

To achieve multi-objective optimization for image super-resolution, we employ evolutionary algorithms to optimize each single-objective optimization subproblem. In each evolutionary step, we select neighboring weight vectors and utilize the information between them to guide the search direction for each single-objective optimization subproblem. Subsequently, we consolidate the optimal solutions from each single-objective optimization subproblem into a single population.

To update the model's weights, we adjust the decomposition weight vectors and neighborhood size based on the distribution and density of solutions in the population. Our method will terminate and provide the Pareto optimal solution set when either the predetermined number of iterations or a specified stop criterion is met. The primary steps of our Multi-Objective Evolutionary Super-Resolution (MOESR) approach are outlined in Algorithm 1:

---

**Algorithm 1** MOESR with Differential Evolution

---

**Input** : Problem: a multi-objective optimization problem, $\min G(x) = (f_1(x), f_2(x), \ldots, f_m(x))^T$;
$T$: the maximum number of iterations;
$L$: the neighborhood size;
$H$: The number of generations each subproblem evolves;
$\lambda_1,...,\lambda_N$:$N$ evenly distributed weight vectors,$\lambda_i = (\lambda_{i1}, \lambda_{i2}, ..., \lambda_{im}), i = 1, 2, ..., N$;
$N$: the number of subproblems.
Subproblems: $F_j(x) = \sum_{i=1}^{m} \lambda_{ji} f_i(x), j = 1, 2, ..., N$
**Output** :A set of non-dominated solutions $EP$.
**Pipeline** : Compute the Euclidean distance between any two weight vectors and find the nearest $L$ weight vectors from each weight vector, for each $i$=1,...,$N$, let $B(i)$={$i_1,...,i_L$} where $\lambda_{i1},...,\lambda_{iL}$ are the $L$ nearest weight vectors to $\lambda_i$;
Generate an initial population population $EP$={$x_1,...,x_N$} by a specific method.
**for** $t = 1$ *to* $T$ **do**
    **for** $i = 1$ *to* $N$ **do**
        For $i$-th subproblem, differential evolution (DE) algorithm is used for evolution, and the initial population is set to $P(i)$={$x_{i1},...,x_{iL}$}, the evolution generation is $H$, and the fitness function is $F_i(x)$. The algorithm output new generation $P^{new}(i)$={$x_{i1}^{new},...,x_{iL}^{new}$}. And the $x_{i1},...,x_{iL}$ in $EP$ is replaced by $x_{i1}^{new},...,x_{iL}^{new}$.

Output the non-dominated solutions $EP$;

---

### 3.3 POPULATION INITIALIZATION

Evolutionary algorithms are a population-based optimization method. In MOESR, the population serves as the initial solution for each sub-problem. However, the traditional random generation of the initial population presents challenges when applied to neural network learning. To tackle this, we propose using pre-trained models as the initial population to expedite the optimization process. To achieve this, it is crucial to employ different pre-trained models for various sub-problems. Consequently, the model must undergo specific fine-tuning for each sub-problem to obtain the initial solution. Nevertheless, optimizing for different sub-problems may not directly influence gradient preferences during model fine-tuning.

To surmount this limitation, we introduce a simple yet effective approach. Commencing the optimization from a well-trained model focused on a single objective effectively represents the trade-off between the two objectives. For the multi-objective super-resolution task, we utilize the L1 loss as the objective for the well-trained model to initialize the optimization process and then optimize further using the SSIM or VGG loss. This strategy enables the intermediate states to strike a balance between the two objectives, resulting in an improved initialization of the population and faster convergence.

### 3.4 DIFFERENTIAL EVOLUTION AND ITS VARIANTS

Optimizing neural network weights is a complex challenge, primarily due to the exceedingly high dimensionality of the problem and the constrained population size for potential solutions. To tackle this complexity, we employ two potent variants of Differential Evolution (DE) aimed at automating the selection of hyperparameters denoted as "f" and "Cr," as well as enriching the repertoire of available strategies. The first of these variants is Self-Adaptive Differential Evolution (SADE), which introduces two pivotal enhancements to enhance DE's performance.

Firstly, SADE incorporates an adaptive crossover rate, CR, drawn from the normal distribution of the previous mean. Secondly, SADE employs a set of four distinct strategies for the weight mutation task. This approach significantly elevates the overall efficiency of the optimization process.

$$\mathbf{v}_{i,g} = \mathbf{x}_{r_1,g} + F \cdot (\mathbf{x}_{r_2,g} - \mathbf{x}_{r_3,g})$$
$$\mathbf{v}_{i,g} = \mathbf{x}_{i,g} + F \cdot (\mathbf{x}_{gbest,g} - \mathbf{x}_{i,g}) + F \cdot (\mathbf{x}_{r_2,g} - \mathbf{x}_{r_3,g})$$
$$\mathbf{v}_{i,g} = \mathbf{x}_{r_1,g} + F \cdot (\mathbf{x}_{r_2,g} - \mathbf{x}_{r_3,g}) + F \cdot (\mathbf{x}_{r_4,g} - \mathbf{x}_{r_5,g})$$
$$\mathbf{v}_{i,g} = \mathbf{x}_{i,g} + \text{rand}_u(0,1) \cdot (\mathbf{x}_{r_1,g} - \mathbf{x}_{i,g}) + F \cdot (\mathbf{x}_{r_2,g} - \mathbf{x}_{r_3,g}).$$

Strategies are chosen based on their historical success probabilities.

Another noteworthy variant is Success-History based Adaptive DE (SHADE). It enhances DE's optimization performance by refining the adjustment of the crossover rate and mutation rate. We use the four strategies from SADE and the historic archive of hyperparameters for each strategy. This cultivates a very rich variety of strategies when optimizing the neural network weight.

## 4 EXPERIMENT

### 4.1 IMPLENTATION DETAILS

**Dataset:** Following (He et al., 2016; Zhang et al., 2018a; Yang et al., 2020; Zhou et al., 2020), we use 800 high-quality (2K resolution) images from the DIV2K dataset (Timofte et al., 2017) as the training set. We evaluate our models on five standard benchmarks: Set5 (Bevilacqua et al., 2012), Set14 (Zeyde et al., 2012), BSD100 (Martin et al., 2001), Urban100 (Huang et al., 2015) and Manga109 (Matsui et al., 2017) in two upscaling factors: $\times 2$, and $\times 4$. All the experiments are conducted with Bicubic (BI) downsampling degradation.

**Evaluation metrics** We extensively employ metrics such as SSIM, PSNR, and LPIPS to evaluate the effectiveness of our model in SR tasks. Additionally, we utilize IGD and HV to represent the model's handling of multi-objective scenarios. Specific details can be found in the Appendix D.

**Training Settings**: We crop the HR patches from the DIV2K dataset (Timofte et al., 2017) for training. Then these patches are downsampled by Bicubic to get the LR patches. For all different downsampling scales in our experiments, we fixed the size of LR patches as $60 \times 60$. All the training patches are augmented by randomly horizontally flipping and rotation of $90°$, $180°$, $270°$ (He et al., 2016). We set the minibatch size to 16 and train our model using the SHADE optimizer and evaluate the impact of different DE optimizer variants in the ablation study. The initial learning rate is set as $10^{-4}$ and then decreases to half for every $2 \times 10^5$ iteration. Training is terminated after $8 \times 10^5$ iterations. All the experiments are implemented on eight NVIDIA 3090 GPUs.

### 4.2 EFFECTIVENESS ON TWO OBJECTIVE METRICS

In this task, we pretrain the model on L1 loss (PSNR) and fine-tune it under the SSIM Loss. We demonstrate the optimization process at different generation phases and evaluate each method using

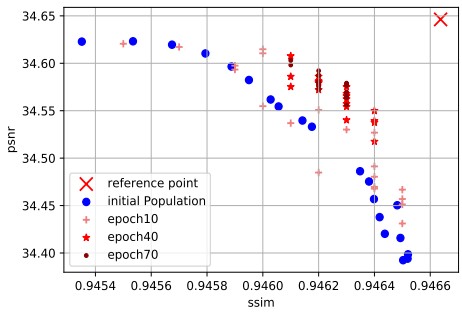 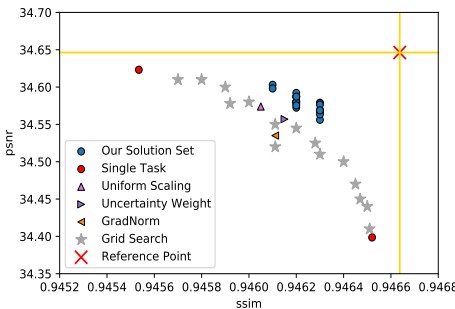

Figure 2: Different generations in MO process.    Figure 3: Different methods in MO problem.

Table 1: **Multi-Object SR Metrics.** This table summarizes the performance of different generations in the MO evolving process.

| Generation | initial | 10th | 40th | 70th |
|---|---|---|---|---|
| GD | 0.00123 | 0.00093 | 0.00071 | 0.00069 |
| HV | 0.00012 | 0.00011 | 0.00006 | 0.00004 |

the multi-objective optimization metric: GD and HV, which proves that our method can well obtain better-optimized solutions for multi-objective problems. As shown in table 1, we could find that the GD to and HV values both show a decreasing trend as the number of iterations increases (generations). The baselines we consider are (i) **uniform scaling**: minimizing a uniformly weighted sum of loss functions $\frac{1}{T}\sum_t L^t$, (ii) **single metric**: solving metrics independently, (iii) **grid search**: exhaustively trying various values from $\{c^t \in [0,1] \mid \sum_t c^t = 1\}$ and optimizing for $\frac{1}{T}\sum_t c^t L^t$, (iv) **Uncertainty Weight:** using the uncertainty weighting proposed by (Kendall et al., 2018), and (v) **GradNorm:** using the normalization proposed by (Chen et al., 2018). We only compare different methods using the EDSR baseline model and the result is shown in Fig. 3. In addition, we compute the GD metric of different method and the result shows that our method performs better in the proposed multi-object SR task, as shown in table 2 It can be observed that, compared to other gradient-based methods, our EA-based approach exhibits a smaller GD, demonstrating the effectiveness and advantages of our method in multi-objective optimization.

Since the pretrained model is trained with L1 loss and prefers the single metric PSNR, we choose a solution with an approximate PSNR value with the original and compare only the SSIM value. We conduct extensive experiments with different models: IMDN (Hui et al., 2019) with lightweight structure, RCAN (Zhang et al., 2018a) with attention mechanisms, EDSR (He et al., 2016) with residual connections, and so on. The results in table 3 show that our MOESR framework significantly improved the SSIM metric, while the PSNR metric remained the same or slightly decreased. Note that EDSR-MO not only achieves an improvement in SSIM metrics on all datasets compared to EDSR, but also achieves some improvement in PSNR metrics on most datasets. In addition, we provide quantitative results to compare the visual perceptual effect of single metric and MO metric, as shown in Fig. 4.

### 4.3 EFFECTIVENESS ON OBJECT-PERCEPTUAL METRICS

To demonstrate the generalizability of our approach, we employ the MOESR method to optimize Objective metrics (PSNR) and Perceptual metrics (LPIPS). As shown in Table 4, we validate our results across five different datasets and four distinct models. We initially use the L1 loss to train our primary model and further fine-tune it using the VGG loss. The results demonstrate that all models

Table 2: **Compare with Other Methods.** This table compares the performance of the different methods in improving the EDSR_baseline model on DIV2K validate dataset. GD is a metric used to evaluate the performance of multi-objective optimization, where lower values indicate better performance.

| | PSNR loss | SSIM loss | Uniform Scaling | mean of Grid Search | Uncertainty Weight | GradNorm | MOESR |
|---|---|---|---|---|---|---|---|
| GD | 0.00121 | 0.00620 | 0.00088 | 0.00094 | 0.00083 | 0.00092 | 0.00069 |

Table 3: Quantitative results in comparison with the state-of-the-art methods. Average PSNR/SSIM for scale factor x2 and x4 on benchmark datasets Set5, Set14, BSD100, Urban100, and Manga109. †denotes the model used our proposed MOESR.

| Method | Scale | Set5 | | Set14 | | BSD100 | | Urban100 | | Manga109 | |
|---|---|---|---|---|---|---|---|---|---|---|---|
| | | PSNR↑ | SSIM↑ | PSNR↑ | SSIM↑ | PSNR↑ | SSIM↑ | PSNR↑ | SSIM↑ | PSNR↑ | SSIM↑ |
| Bicubic | x2 | 33.66 | 0.9299 | 30.24 | 0.8688 | 29.56 | 0.8431 | 26.88 | 0.8403 | 30.8 | 0.9339 |
| IMDN | x2 | 38.00 | 0.9594 | 33.47 | 0.9159 | 32.09 | 0.8996 | 32.17 | 0.9283 | 38.42 | 0.9784 |
| IMDN† | x2 | 37.67 | 0.9615 | 33.46 | 0.9198 | 32.07 | 0.9027 | 32.02 | 0.9302 | 38.54 | 0.9861 |
| EDSR-baseline | x2 | 37.99 | 0.9604 | 33.57 | 0.9175 | 32.16 | 0.8994 | 31.98 | 0.9272 | 38.42 | 0.9769 |
| EDSR-baseline† | x2 | 37.83 | 0.9608 | 33.55 | 0.9190 | 32.08 | 0.9030 | 31.90 | 0.929 | 38.22 | 0.9789 |
| EDSR | x2 | 38.11 | 0.9602 | 33.92 | 0.9195 | 32.32 | 0.9013 | 32.93 | 0.9351 | 39.14 | 0.9773 |
| EDSR† | x2 | 38.19 | 0.9606 | 33.939 | 0.9213 | 32.42 | 0.9024 | 32.964 | 0.9361 | 39.11 | 0.9787 |
| RCAN | x2 | 38.27 | 0.9614 | 34.12 | 0.9216 | 32.41 | 0.9027 | 33.34 | 0.9384 | 39.44 | 0.9786 |
| RCAN† | x2 | 38.23 | 0.9634 | 34.10 | 0.9229 | 32.36 | 0.9066 | 33.28 | 0.9399 | 39.33 | 0.9817 |
| Bicubic | x4 | 33.66 | 0.9299 | 30.24 | 0.8688 | 29.56 | 0.8431 | 26.88 | 0.8403 | 30.8 | 0.9339 |
| IMDN | x4 | 32.21 | 0.8948 | 28.58 | 0.7811 | 27.56 | 0.7353 | 26.04 | 0.7838 | 30.35 | 0.9075 |
| IMDN† | x4 | 31.97 | 0.9050 | 28.43 | 0.7854 | 27.44 | 0.7412 | 25.90 | 0.7889 | 30.26 | 0.9087 |
| EDSR_baseline | x4 | 32.10 | 0.8938 | 28.58 | 0.7813 | 27.57 | 0.7357 | 26.04 | 0.7849 | 30.35 | 0.9067 |
| EDSR-baseline† | x4 | 32.07 | 0.8970 | 28.56 | 0.7831 | 27.56 | 0.7377 | 26.02 | 0.7878 | 30.33 | 0.9111 |
| EDSR | x4 | 32.46 | 0.8968 | 28.80 | 0.7876 | 27.71 | 0.7420 | 26.64 | 0.8033 | 31.02 | 0.9148 |
| EDSR† | x4 | 32.47 | 0.8998 | 28.79 | 0.7883 | 27.707 | 0.7434 | 26.63 | 0.8054 | 30.98 | 0.9169 |
| RCAN | x4 | 32.63 | 0.9002 | 28.87 | 0.7889 | 27.77 | 0.7436 | 26.82 | 0.8087 | 31.22 | 0.9173 |
| RCAN† | x4 | 32.62 | 0.9103 | 28.84 | 0.7967 | 27.50 | 0.7460 | 26.78 | 0.8096 | 31.15 | 0.9189 |

Table 4: Quantitative results in comparison with the state-of-the-art methods. Average PSNR/LPIPS for scale factor x2 and x4 on benchmark datasets Set5, Set14, BSD100, Urban100, and Manga109. †denotes the model used our proposed MOESR.

| Method | Scale | Set5 | | Set14 | | BSD100 | | Urban100 | | Manga109 | |
|---|---|---|---|---|---|---|---|---|---|---|---|
| | | PSNR↑ | LPIPS↓ | PSNR↑ | LPIPS↓ | PSNR↑ | LPIPS↓ | PSNR↑ | LPIPS↓ | PSNR↑ | LPIPS↓ |
| IMDN | x2 | 38.00 | 0.0459 | 33.47 | 0.0727 | 32.09 | 0.0742 | 32.17 | 0.0556 | 38.42 | 0.0211 |
| IMDN† | x2 | 37.86 | 0.0440 | 33.45 | 0.0712 | 32.09 | 0.0726 | 32.13 | 0.0527 | 38.41 | 0.0205 |
| EDSR-baseline | x2 | 37.99 | 0.0456 | 33.57 | 0.0715 | 32.16 | 0.073 | 31.98 | 0.0538 | 38.42 | 0.0212 |
| EDSR-baseline† | x2 | 37.84 | 0.0435 | 33.57 | 0.0714 | 32.13 | 0.0725 | 31.94 | 0.0529 | 38.37 | 0.0201 |
| EDSR | x2 | 38.11 | 0.0440 | 33.92 | 0.0691 | 32.32 | 0.0713 | 32.93 | 0.0487 | 39.14 | 0.0202 |
| EDSR† | x2 | 38.09 | 0.0431 | 33.91 | 0.0679 | 32.29 | 0.0701 | 32.91 | 0.0481 | 39.11 | 0.0194 |
| RCAN | x2 | 38.27 | 0.0436 | 34.12 | 0.0689 | 32.41 | 0.0713 | 33.34 | 0.0485 | 39.44 | 0.0201 |
| RCAN† | x2 | 38.24 | 0.0426 | 34.10 | 0.0673 | 32.36 | 0.0705 | 33.31 | 0.0476 | 39.39 | 0.0193 |
| IMDN | x4 | 32.21 | 0.1041 | 28.58 | 0.1577 | 27.56 | 0.1612 | 26.04 | 0.1465 | 30.35 | 0.0718 |
| IMDN† | x4 | 31.96 | 0.1032 | 28.49 | 0.1562 | 27.49 | 0.1601 | 25.96 | 0.1457 | 30.26 | 0.0704 |
| EDSR_baseline | x4 | 32.10 | 0.1049 | 28.58 | 0.1563 | 27.57 | 0.1593 | 26.04 | 0.1593 | 30.35 | 0.0697 |
| EDSR-baseline† | x4 | 32.07 | 0.1037 | 28.55 | 0.1549 | 27.56 | 0.1564 | 26.01 | 0.1512 | 30.31 | 0.0674 |
| EDSR | x4 | 32.46 | 0.1017 | 28.80 | 0.1506 | 27.71 | 0.1543 | 26.64 | 0.1264 | 31.02 | 0.0648 |
| EDSR† | x4 | 32.47 | 0.1020 | 28.79 | 0.1134 | 27.69 | 0.1120 | 26.63 | 0.1027 | 30.98 | 0.0639 |
| RCAN | x4 | 32.63 | 0.0691 | 28.87 | 0.0829 | 27.77 | 0.0944 | 26.82 | 0.0815 | 31.22 | 0.0638 |
| RCAN† | x4 | 32.61 | 0.0682 | 28.85 | 0.0817 | 27.60 | 0.0910 | 26.78 | 0.0803 | 31.19 | 0.0624 |

utilizing our MOESR method achieve significant improvements in LPIPS across all datasets. It is evident that our method not only enhances the effectiveness of LPIPS over the original model but also preserves a substantial portion of the PSNR values. This experimental outcome substantiates that our method represents a superior trade-off. In summary, it confirms the broad applicability of our approach and its proficiency in addressing both objective and perceptual metrics.

## 4.4 ABLATION STUDY

**Effectiveness of each component**. We conducted the ablation study on Urban100 and Mangna109 to evaluate the contribution of each component in our methodology. This includes the population initialization method and the effectiveness of Differential Evolution and its variants. As shown in Table .5, *Random* refers to adding random noise to a pretrained model to generate the parent set. And *L1* represents the use of L1 loss for further training of the pretrained model to procure the parent set. Our findings indicate that both PSNR and SSIM yield superior performance when our proposed method is used for the initial parent set. This strongly corroborates the effectiveness of our proposed population initialization method.

In Table 6, we compare the effects of three commonly employed variants of the DE method. We observed that both PSNR and SSIM demonstrate a rising trend commensurate with the increasing complexity of the DE method. Notably, the SHADE variant delivered the most optimal results.

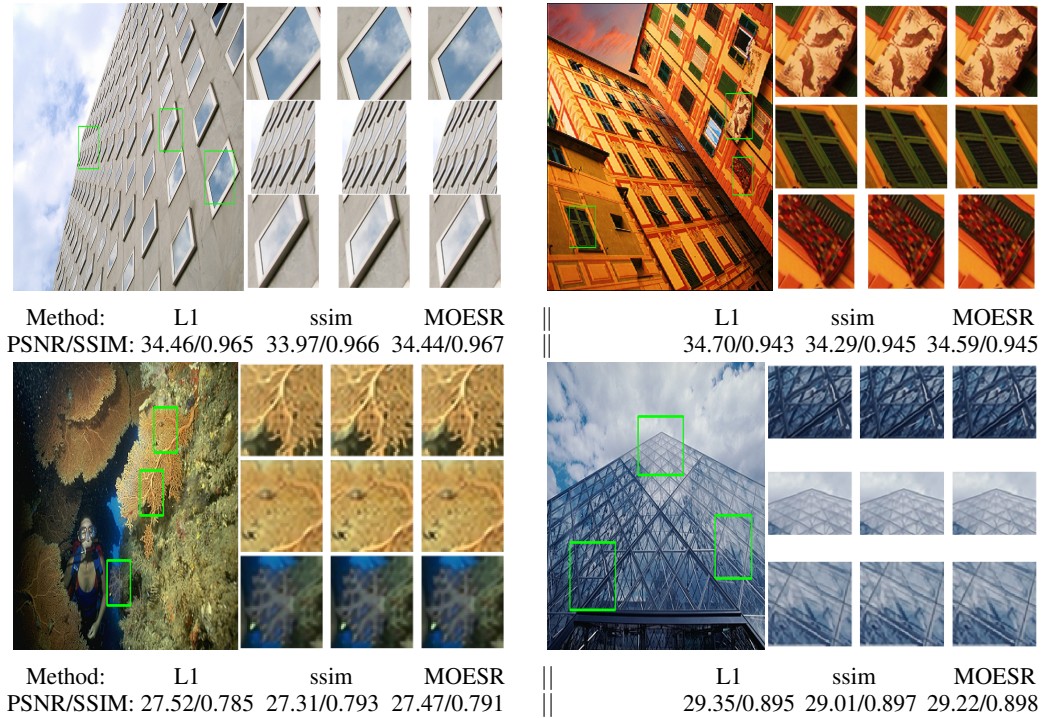

| Method: | L1 | ssim | MOESR | || | L1 | ssim | MOESR |
|---|---|---|---|---|---|---|---|
| PSNR/SSIM: | 34.46/0.965 | 33.97/0.966 | 34.44/0.967 | || | 34.70/0.943 | 34.29/0.945 | 34.59/0.945 |

| Method: | L1 | ssim | MOESR | || | L1 | ssim | MOESR |
|---|---|---|---|---|---|---|---|
| PSNR/SSIM: | 27.52/0.785 | 27.31/0.793 | 27.47/0.791 | || | 29.35/0.895 | 29.01/0.897 | 29.22/0.898 |

Figure 4: Visual results with Bicubic downsampling (x2) on images from BSD100 and Urban100. Our proposed MOESR provides a better trade-off result.

Table 5: Image restoration quality of RCANx2 under different population initialization methods.

| Datasets | Metric | Random | L1 | Ours |
|---|---|---|---|---|
| BSD100 | PSNR↑ | 33.25 | 32.41 | 32.36 |
| | SSIM↑ | 0.9375 | 0.9379 | 0.9066 |
| Mangna109 | PSNR↑ | 39.31 | 39.39 | 39.33 |
| | SSIM↑ | 0.9702 | 0.9783 | 0.9817 |

Table 6: Image restoration quality of RCANx2 under Differential Evolution and its variants.

| Datasets | Metric | DE | SHDE | SHADE |
|---|---|---|---|---|
| BSD100 | PSNR↑ | 32.23 | 32.27 | 32.36 |
| | SSIM↑ | 0.9042 | 0.9051 | 0.9066 |
| Mangna109 | PSNR↑ | 39.16 | 39.25 | 39.33 |
| | SSIM↑ | 0.9797 | 0.9805 | 0.9817 |

## 5    CONCLUSION

In conclusion, this stduy introduces the Multi-Objective Evolutionary Algorithm for Image Super-Resolution (MOESR), a comprehensive framework designed to overcome the limitations of existing deep learning methods in super-resolution. By addressing multiple objectives in image enhancement, MOESR offers a balanced optimization approach, moving beyond the sole focus on peak signal-to-noise ratio (PSNR) improvement. Our method decomposes the problem into sub-problems and utilizes a novel evolutionary algorithm to generate an initial population, with improved mutation, crossover, and update processes via an enhanced differential evolution algorithm. MOESR outperforms gradient-based methods by eliminating the need for gradient calculations for each objective, thereby mitigating issues like gradient vanishing and local optima. Moreover, it boasts lower computational complexity, particularly beneficial for high-dimensional problems and deep networks. Extensive experiments validate MOESR's promising performance across benchmarks and multi-objective tasks, offering a groundbreaking capability to balance objective and perceptual metrics in image quality. MOESR is poised to advance image super-resolution and multi-objective optimization research.

**Limitation and Future work**    The development of multiple objective evaluation metrics is a complex endeavor due to the significant variations in optimization tasks. Nevertheless, these metrics play a crucial role in guiding multi-objective optimization processes. In the realm of Super-Resolution (SR) tasks, research in this particular aspect has been relatively limited. The evaluation metrics we have selected suffer from insufficient theoretical analysis and empirical evidence, necessitating further investigation. Additionally, it is imperative to conduct deeper studies on improved population initialization methods.

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
