# OpenReview forum: "MOESR: MULTI-OBJECTIVE EVOLUTIONARY ALGORITHM FOR IMAGE SUPER-RESOLUTION"
_ICLR.cc/2024/Conference — Submitted to ICLR 2024_

### Official Review · Reviewer_rNa3 · 2023-10-31

**Soundness:** 3 good
**Presentation:** 2 fair
**Contribution:** 2 fair
**Rating:** 5
**Confidence:** 5

**Summary:**

The authors propose a general Pareto efficient algorithmic framework for multi-objective SISR. The framework is designed  to be adaptable to various models and objectives. Based on the MOESR, the authors show the way to generate the Pareto Frontier and establish the evaluation metrics for SR tasks, which is claimed as the first work to select a fair SR solution from Pareto Frontier using appropriate fairness metrics. The authors firstly propose an enhanced version of differential evolution algorithm for multi-objective super resolution task. The results on the benchmark datasets show that the algorithm performs better than previous SOTA methods.

**Strengths:**

1. The paper demonstrate the multi-objective evolutionary SR framework for multi-objective SR task. This is the first time using evolutionary strategy to improve the multi-objective SR task to my knowledge.

2. The paper is easy to follow.

3. The ablation study of the paper is sufficient.

**Weaknesses:**

1. To my knowledge, the multi objective loss is widely implemented in the industry. The motivation of the paper needs to be re-organized. And the first contribution of the paper is kind of overclaimed and needs to be re-organized.

2. The methods in the Table 3 are not the most recently published though they are all very classic methods. What is the performance of the MOESR on the transformer based methods like SwinIR?

3. The most confusing part is the visualization. I cannot tell the difference on Figure 1 and Figure 4. Please use other images to visualize.

**Questions:**

Please see the weakness.

---

### Official Review · Reviewer_omD1 · 2023-11-01

**Soundness:** 1 poor
**Presentation:** 2 fair
**Contribution:** 1 poor
**Rating:** 1
**Confidence:** 4

**Summary:**

The paper proposes the use of multiobjective evolutionary algorithm to optimize the weights of neural networks used for producing super-resolution version of images. The paper argue and proposes an Pareto-based multiobjective optimization joined to a evolutionary algorithm used to optimize neural network weights as an approach to achieve good performance over several measures. The approach is tested over several models on some image datasets.

**Strengths:**

- The paper aims at proposed a novel approach to super-resolution based on multi-objective optimization.
- Results over various datasets and models.

**Weaknesses:**

- The core justifications of the approach doesn’t hold track, or at least are not well justified. The use of Pareto-based multiobjective optimization usually implies objectives that are contradictory, here the objectives aim at similar goals, although they may not be perfectly aligned. The need for such MO is not clear to me.
- The justification of gradient-free optimization is not clear. These optimization approaches are much weaker and would require more computation to achieve strong results than gradient-based optimization. We currently have a model and losses from which we can get derivatives, I don’t see the issue in using them, the justifications provided are not convincing.
- The results provided are showing very small improvements regarding the values of PSNR and SSIM (e.g., Fig. 3), even if it looks like a Pareto front, I think the differences obtained are just a slight misalignment between the two metrics, which overall aims at a similar goal. Moreover, at such levels, the differences are not necessarily meaningful, especially for the perceptual losses, which are not to be interpreted to be super precise. The gains provided by the proposed approach looks rather like to me a statistical flukes coming from the fact that we are dealing with population-based optimization, and as such we are able to pick a set of solutions in the population that appears to behave better on our train/validation/test sets (not sure which one was used in Fig. 3), but are in fact within statistical error margins.
- Looking at Figure 3, with the different images, it is not clear what is what, but overall it is very difficult to figure out anything from the images, they look quite similar and overall not of a very good resolution. Having clearly identified low resolution, high resolution of the patch as well with the one generated with some of the approaches would be helpful. A good example of figures showing well the difference are the on in the paper “A Deep Journey into Super-resolution: A Survey” (https://arxiv.org/pdf/1904.07523.pdf), where differences between the approaches evaluated are obvious (see Figs. 6 and 7 as great examples).

**Questions:**

I think the greatest issues of the paper lies in its motivations. The fact we should use Pareto MO is really not clear neither well justified. The natural reflex is to combine several objectives in the optimization loss with some weighting (something in the form L = f + \lambda g, where \lambda is the weight for the tradeoff between f and g objectives in the loss)  seems the way to go at first. The paper proposes to use something similar, but the adjustment of the \lambda appears not proper to me, the scale of the two measures are quite different, and various scales of \lambda need to be tested. Moreover, the use of Pareto MO usually comes from the fact that the objectives at hand working in some sort of opposition, improving one value should lead to a decrease of the others, and some on. I don’t think this is the case in the current case, as the losses are mostly aiming for the same goal, with measurements on different aspects and a slightly different misalignment, but nothing strong. It makes much more sense to combine them than to optimize them as opposed objectives.

Also, the use of evolutionary algorithms is justified only by the fact that population-based optimization is compatible with Pareto MO, where we are dealing with a set of non-dominated solutions, rather than just the single current best solution (as we usually have with gradient-based optimization). But as soon as we are challenging the notion of making use of Pareto MO, the need for evolutionary algorithm vs gradient-descent does not hold. In brief, the need for Pareto MO in the current context should be made in a more convincing manner.

The results reported show small gains over the objectives values. Moreover, the methodology to pick the best solution for Pareto MO optimization is not clear. How the solutions reported in Tables 3 and 4 picked? Are they chosen from the Pareto set (non-dominated solutions) according to their test performances (pick the best on test), or on some other training or validation set? Because picking them on the same dataset that the one used to report the results is not fair, as we will pick one that may be “by chance” performing well, given there should be some random variability observed with these measures, compared to the single solutions provided by the gradient approach. That should be explained further to insure that the gains observed with the proposed approach are not just statistical flukes.

**Details Of Ethics Concerns:**

No ethical concerns with this paper.

---

### Official Review · Reviewer_EdLo · 2023-11-02

**Soundness:** 2 fair
**Presentation:** 2 fair
**Contribution:** 2 fair
**Rating:** 5
**Confidence:** 3

**Summary:**

In this paper, the authors proposed a general Pareto efficient algorithmic framework (MOESR) for multi-objective image super-resolution. The proposedd method decomposes the SR problem into sub-problems and utilizes an evolutionary algorithm to generate an initial population. The authors have conducted experiments to demonstrate the effectiveness of their method.

**Strengths:**

1. The motivation of this paper is clear.
2. The logic flow is easy to follow.

**Weaknesses:**

1. It has been proved in [C1] that there is a natural tradeoff between SR accuracy (in terms of PSNR) and perceptual quality (in terms of LPIPS).
[C1] The perception-distortion tradeoff, CVPR 2018.

2. The authors claimed that they propose the first method capable of simultaneously addressing both objective and perceptual metrics. However, there are several work that have studied this issue. The authors should carefully review the existing studies and discuss the original contributions upon them.
[C2] Wavelet domain style transfer for an effective perception-distortion tradeoff in single image super-resolution, ICCV 2019.
[C3] Perception-distortion balanced ADMM optimization for single-image super-resolution, ECCV 2022.

3. Typos:
(1) In Sec.1: Super-resolution (SR) is a extensively studied field --> an extensively
(2) In Sec.4.1: as shown in table 2 It can be observed that --> as shown in Table 2. It can be observed that

**Questions:**

1. In Table 1, IMDN equipped with the proposed method can simultaneously achieve higher PSNR and SSIM scores on the Manga109 dataset. similar results can be also observed in EDSR on the Urban100 dataset. More explanations should be given.

---

### Official Review · Reviewer_AESC · 2023-11-12

**Soundness:** 2 fair
**Presentation:** 3 good
**Contribution:** 2 fair
**Rating:** 3
**Confidence:** 4

**Summary:**

This paper introduces a novel approach for the super resolution task, aiming to learn strategies that benefit multiple evaluation metrics. The proposed method, called MOESR (Multi-Multi-Objective Evolutionary Algorithm for Image Super-Resolution), is extensively evaluated across various benchmarks and multi-objective tasks. The experimental results validate the promising performance of MOESR.

**Strengths:**

The paper is generally well-written and easy to follow. The authors have illustrated their motivations using bullet points to provide a clear understanding of their objectives.

**Weaknesses:**

The primary concern raised is the lack of necessity for employing multi-objective optimization in the image super-resolution task. It is pointed out that there are no conflicts observed between different evaluation metrics, which is a crucial aspect of multi-objective optimization. Furthermore, the experiments conducted did not demonstrate much advantage over the baseline method. Additionally, it is noted that there is a computational overhead when using eight 3090 GPUs.

**Questions:**

Please refer to the weakness.

---

### Meta-Review · Area_Chair_rmGU · 2023-12-12

**Metareview:**

The paper is generally well-written and easy to follow (e.g., logic flow) with clear motivations.

The lack of necessity for employing multi-objective optimization in the image super-resolution task weakens the paper. The experiments can hardly demonstrate much advantage over the baselines. A natural tradeoff between SR accuracy (in terms of PSNR) and perceptual quality (in terms of LPIPS) has been extensively investigated before. The authors claimed that they propose the first method capable of simultaneously addressing both objective and perceptual metrics. However, several works have studied this issue.

All reviewers give negative scores. The authors did not respond.

**Justification For Why Not Higher Score:**

N/A

**Justification For Why Not Lower Score:**

N/A

---

### Decision · Program_Chairs · 2024-01-16

Reject